# PointDAN: A Multi-Scale 3D Domain Adaption Network for Point Cloud Representation

[1]**Can Qin,**[*] [2]**Haoxuan You,**[*] [1]**Lichen Wang,** [3]**C.-C. Jay Kuo,** [1,4]**Yun Fu**

[1]Department of Electrical & Computer Engineering, Northeastern University
[2]Department of Computer Science, Columbia University
[3]Department of Electrical and Computer Engineering, University of Southern California
[4]Khoury College of Computer Science, Northeastern University
`qin.ca@husky.neu.edu, haoxuan.you@columbia.edu,`
`wanglichenxj@gmail.com, cckuo@sipi.usc.edu, yunfu@ece.neu.edu`

## Abstract

Domain Adaptation (DA) approaches achieved significant improvements in a wide range of machine learning and computer vision tasks (*i.e.*, classification, detection, and segmentation). However, as far as we are aware, there are few methods yet to achieve domain adaptation directly on 3D point cloud data. The unique challenge of point cloud data lies in its abundant spatial geometric information, and the semantics of the whole object is contributed by including regional geometric structures. Specifically, most general-purpose DA methods that struggle for global feature alignment and ignore local geometric information are not suitable for 3D domain alignment. In this paper, we propose a novel 3D Domain Adaptation Network for point cloud data (PointDAN). PointDAN jointly aligns the global and local features in multi-level. For local alignment, we propose Self-Adaptive (SA) node module with an adjusted receptive field to model the discriminative local structures for aligning domains. To represent hierarchically scaled features, node-attention module is further introduced to weight the relationship of SA nodes across objects and domains. For global alignment, an adversarial-training strategy is employed to learn and align global features across domains. Since there is no common evaluation benchmark for 3D point cloud DA scenario, we build a general benchmark (*i.e.*, PointDA-10) extracted from three popular 3D object/scene datasets (*i.e.*, ModelNet, ShapeNet and ScanNet) for cross-domain 3D objects classification fashion. Extensive experiments on PointDA-10 illustrate the superiority of our model over the state-of-the-art general-purpose DA methods.[1]

## 1 Introduction

3D vision has achieved promising outcomes in wide-ranging real-world applications (*i.e.*, autonomous cars, robots, and surveillance system). Enormous amounts of 3D point cloud data is captured by depth cameras or LiDAR sensors nowadays. Sophisticated 3D vision and machine learning algorithms are required to analyze its content for further exploitation. Recently, the advent of Deep Neural Network (DNN) has greatly boosted the performance of 3D vision understanding including tasks of classification, detection, and segmentation[22, 9, 37, 41]. Despite its impressive success, DNN requires massive amounts of labeled data for training which is time-consuming and expensive to collect. This issue significantly limits its promotion in the real world.

Domain adaptation (DA) solves this problem by building a model utilizing the knowledge of label-rich dataset, *i.e.*, source domain, which generalizes well on the label-scarce dataset, *i.e.*, target domain.

[*]Equal Contribution.

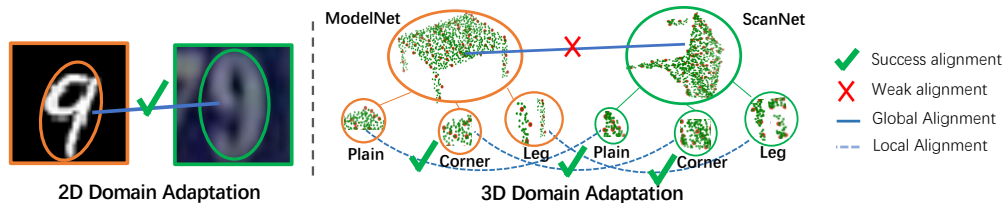

Figure 1: Comparison between 2D-based and 3D-based DA approaches.

However, due to the shifts of distribution across different domains/datasets, a model trained on one domain usually performs poorly on other domains. Most DA methods address this problem by either mapping original features into a shared subspace or minimizing instance-level distances, such as MMD, CORAL *etc.*, to mix cross-domain features [2, 18, 31]. Currently, inspired by Generative Adversarial Network (GAN) [12], adversarial-training DA methods, like DANN, ADDA, MCD *etc.*, have achieved promising performance in DA and drawn increasing attentions [10, 32, 26]. They deploy a zero-sum game between a discriminator and a generator to learn domain-invariant representations. However, most of the existing DA approaches mainly target on 2D vision tasks, which globally align the distribution shifts between different domains. While for 3D point cloud data, the geometric structures in 3D space can be detailedly described, and different local structures also have clear semantic meaning, such as legs for chairs, which in return combine to form the global semantics for a whole object. As shown in Fig. 1, two 3D objects might be weak to align in global, but would have similar 3D local structures, which are easier to be aligned. So it is urgently desired for a domain adaptation framework to focus on local geometric structures in 3D DA scenario.

To this end, this paper introduces a novel point-based Unsupervised Domain Adaptation Network (PointDAN) to achieve unsupervised domain adaptation (UDA) for 3D point cloud data. The key to our approach is to jointly align the multi-scale, *i.e.*, global and local, features of point cloud data in an end-to-end manner. Specifically, the Self-Adaptive (SA) nodes associated with an adjusted receptive field are proposed to dynamically gather and align local features across domains. Moreover, a node attention module is further designed to explore and interpret the relationships between nodes and their contributions in alignment. Meanwhile, an adversarial-training strategy is deployed to globally align the global features. Since there are few benchmarks for DA on 3D data ( *i.e.*, point cloud) before, we build a new benchmark named PointDA-10 dataset for 3D vision DA. It is generated by selecting the samples in 10 overlapped categories among three popular datasets (*i.e.*, ModelNet [35], ShapeNet [3] and ScanNet [5]). In all, the contributions of our paper could be summarized in three folds:

- We introduce a novel 3D-point-based unsupervised domain adaptation method by locally and globally align the 3D objects' distributions across different domains.
- For local feature alignment, we propose the Self-Adaptive (SA) nodes with a node attention to utilize local geometric information and dynamically gather regional structures for aligning local distribution across different domains.
- We collect a new 3D point cloud DA benchmark, named PointDA-10 dataset, for fair evaluation of 3D DA methods. Extensive experiments on PointDA-10 demonstrate the superiority of our model over the state-of-the-art general-purpose DA methods.

## 2   Related Works

### 2.1   3D Vision Understanding

Different from 2D vision, 3D vision has various data representation modalities: multi-view, voxel grid, 3D mesh and point cloud data. Deep networks have been employed to deal with the above different formats of 3D data [29, 19, 36, 8]. Among the above modalities, point cloud, represented by a set of points with 3D coordinates $\{x, y, z\}$, is the most straightforward representation to preserve 3D spatial information. Point cloud can be directly obtained by LiDAR sensors, which brings a lot of 3D environment understanding applications from scene segmentation to automatic driving. PointNet [22] is the first deep neural networks to directly deal with point clouds, which proposes a symmetry function and a spatial transform network to obtain the invariance to point permutation. However,

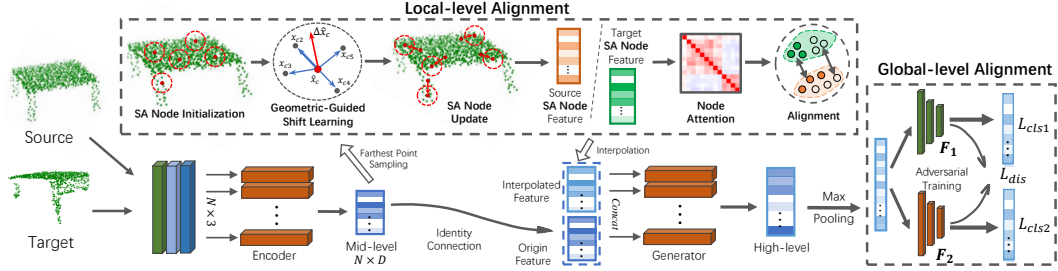

Figure 2: Illustration of PointDAN which mainly consists of local-level and global-level alignment.

local geometric information is vital for describing object in 3D space, which is ignored by PointNet. So recent work mainly focuses on how to effectively utilize local feature. For instance, in PointNet++ [23], a series of PointNet structures are applied to local point sets with varied sizes and local features are gathered in a hierarchical way. PointCNN [17] proposes $\chi$-Conv to aggregate features in local pitches and applies a bottom-up network structure like typical CNNs. In 3D object detection tasks, [41] proposes to divide a large scene into many voxels, where features of inside points are extracted respectively and a 3D Region Proposal Network (RPN) structure is followed to obtain detection prediction.

In spite of the broad usage, point cloud data has significant drawbacks in labeling efficiency. During labeling, people need to rotate several times and look through different angles to identify an object. In real-world environment where point cloud data are scanned from LiDAR, it also happens that some parts are lost or occluded (*e.g.* tables lose legs), which makes efficient labeling more difficult. Under this circumstance, a specific 3D point-based unsupervised domain adaptation method designed to mitigate the domain gap of source labeled data and target unlabeled data is extremely desired.

## 2.2 Unsupervised Domain Adaptation (UDA)

The main challenge of UDA is that distribution shift (*i.e.,* domain gap) exists between the target and source domain. It violates the basic assumption of conventional machine learning algorithms that training samples and test samples sharing the same distribution. To bridge the domain gap, UDA approaches match either the marginal distributions [30, 21, 11, 33] or the conditional distributions [38, 4] between domains via feature alignment. It addresses this problem by learning a mapping function $f$ which projects the raw image features into a shared feature space across domains. Most of them attempt to maximizing the inter-class discrepancy, while minimize the intra-class distance in a subspace simultaneously. Various methods, such as Correlation Alignment (CORAL) [31], Maximum Mean Discrepancy (MMD) [2, 18], or Geodesic distance [13] have been proposed.

Apart from the methods aforementioned, many DNN-based domain adaptation methods have been proposed due to their great capacity in representation learning [14, 28, 16]. The key to these methods is to apply DNN to learn domain-invariant features through an end-to-end training scenario. Another kind of approach utilizes adversarial training strategy to obtain the domain invariant representations [10, 32, 7, 24]. It includes a discriminator and a generator where the generator aims to fool the discriminator until the discriminator is unable to distinguish the generated features between the two domains. Such approaches include Adversarial Discriminative Domain Adaptation (ADDA) [32], Domain Adversarial Neural Network (DANN) [10], Maximum Classifier Discrepancy (MCD) [26] *etc.*

Most of UDA methods are designed for 2D vision tasks and focus on the alignment of global image features across different domains. While in 3D data analytical tasks, regional and local geometry information is crucial for achieving good learning performance. Zhou et al. [40] firstly introduced UDA on the task of 3D keypoint estimation relying on the regularization of multi-view consistency term. However, this method cannot be extended to more generalized tasks, *i.e.*, classification. In [27, 34], point cloud data are first projected into 2D images (bird-eye view or front view), and 2D DA methods are applied, which would lose essential 3D geometric information. To this end, we propose a generalized 3D point-based UDA framework. It well preserves the local structures and explores the

global correlations of all local features. Adversarial training strategies are further employed to locally and globally align the distribution shifts across the source and target domains.

## 3 Proposed Model

### 3.1 Problem Definition and Notation

In 3D point-based UDA, we have the access to labeled source domain $\mathcal{S} = \{(\mathbf{x}_i^s, y_i^s)\}_{i=1}^{n_s}$ where $y_i^s \in \mathcal{Y} = \{1, ..., Y\}$ with $n_s$ annotated pairs and target domain $\mathcal{T} = \{\mathbf{x}_j^t\}_{j=1}^{n_t}$ of $n_t$ unlabeled data points. The inputs are point cloud data usually represented by 3-dimensional coordinates $(x, y, z)$ where $\mathbf{x}_i^s, \mathbf{x}_j^t \in \mathcal{X} \subset \mathbb{R}^{T \times 3}$, where $T$ is the number of sampling points of one 3D object, with the same label space $\mathcal{Y}_s = \mathcal{Y}_t$. It is further assumed that two domains are sampled from the distributions $P_s(\mathbf{x}_i^s, y_i^s)$ and $P_t(\mathbf{x}_i^t, y_i^t)$ respectively while the i.i.d. assumption is violated due to the distribution shift $P_s \neq P_t$. The key to UDA is to learn a mapping function $\Phi : \mathcal{X} \rightarrow \mathbb{R}^d$ that projects raw inputs into a shared feature space $\mathcal{H}$ spreadable for cross-domain samples.

### 3.2 Local Feature Alignment

The local geometric information plays an important role in describing point cloud objects as well as domain alignment. As illustrated in Fig. 1, given the same "table" class, the one from ScanNet misses parts of legs due to the obstacles through LiDAR scanning. The key to align these two "tables" is to extract and match the features of similar structures, *i.e.,* plains, while ignoring the different parts. To utilize the local geometric information, we propose to adaptively select and update key nodes for better fitting the local alignment.

**Self-Adaptive Node Construction:** Here we give the definition of *node* in point cloud. For each point cloud, we represent its $n$ local geometric structures as $n$ point sets $\{S_c | S_c = \{\hat{x}_c, x_{c1}, ..., x_{ck}\}, x \subseteq \mathbb{R}^3\}_{c=1}^n$, where the $c$-th region $S_c$ contains a node $\hat{x}_c$ and its surrounding $k$ nearest neighbor points $\{x_{c1}, ..., x_{ck}\}$. The location of a node decides where the local region is and what points are included.

To achieve local features, directly employing the farthest point sampling or random sampling to get the center node is commonly used in previous work [23, 17]. These methods guarantee full coverage over the whole point cloud. However, for domain alignment, it is essential to make sure that these nodes cover the structures of common characteristics in 3D geometric space and drop the parts unique to certain objects. In this way, the local regions sharing similar structures are more proper to be aligned, while the uncommon parts would bring a negative transfer influence.

Inspired by deformable convolution in 2D vision [6], we propose a novel geometric-guided shift learning module, which makes the input nodes self-adaptive in receptive field for network. Different from Deformable Convolution where semantic features are used for predicting offset, we utilize the local edge vector as a guidance during learning. As show in Fig. 2, our module transforms semantic information of each edge into its weight and then we aggregate the weighted edge vectors together to obtain our predicted offset direction. Intuitively, the prediction shift is decided by the voting of surrounding edges with different significance. We first initialize the location of node by the farthest point sampling over the point cloud to get $n$ nodes, and their $k$ nearest neighbor points are collected together to form $n$ regions. For the $c$-th node, its offset is computed as:

$$\Delta\hat{x}_c = \frac{1}{k}\sum_{j=1}^{k}(R_T(\mathbf{v}_{cj} - \hat{\mathbf{v}}_c) \cdot (x_{cj} - \hat{x}_c)), \tag{1}$$

where $\hat{x}$ and $x_{cj}$ denote location of node and its neighbor point, so $x_{cj} - \hat{x}_c$ means the edge direction. $\mathbf{v}_{cj}$ and $\hat{\mathbf{v}}_c$ are their mid-level point feature extracted from the encoder $\mathbf{v} = E(x|\Theta_E)$ and $R_T$ is the weight from one convolution layer for transforming feature. We apply the bottom 3 feature extraction layers of PointNet as the encoder $E$. $\Delta\hat{x}_c$ is the predicted location offset of the $c$-th node.

After obtaining learned shift $\Delta\hat{x}_c$, we achieve the self-adaptive update of nodes and their regions by adding shift back to node $\hat{x}_c$ and finding their $k$ nearest neighbor points:

$$\hat{x}_c = \hat{x}_c + \Delta\hat{x}_c, \tag{2}$$

$$\{x_{c1}, ..., x_{ck}\} = kNN(\hat{x}_c | x_j, j = 0, ..., M - 1). \tag{3}$$

Then the final node features $\hat{\mathbf{v}}_c$ is computed by gathering all the point features inside their regions:

$$\hat{\mathbf{v}}_c = \max_{j=1,..,k} R_G(\mathbf{v}_{cj}). \tag{4}$$

where $R_G$ is the weight of one convolution layer for gathering point features in which $R_G \bigcup R_T = \mathcal{R}$, and the output node features are employed for local alignment. For better engaging SA node features, we also interpolate them back into each point following the interpolation strategy in [23] and fuse them with the original point features from a skip connection. The fused feature is input into next-stage generator for higher-level processing.

**SA Node Attention:** Even achieving SA nodes, it is unreasonable to assume that every SA node contributes equally to the goal of domain alignment. The attention module, which is designed to model the relationship between nodes, is necessary for weighting the contributions of different SA nodes for domain alignment and capturing the features in larger spatial scales. Inspired by the channel attention [39], we apply a node attention network to model the contribution of each SA nodes for alignment by introducing a bottleneck network with a residual structure [14]:

$$\mathbf{h}_c = \varphi(W_U \delta(W_D \mathbf{z}_c)) \cdot \hat{\mathbf{v}}_c + \hat{\mathbf{v}}_c, \tag{5}$$

where $\mathbf{z}_c = \mathbb{E}(\hat{\mathbf{v}}_c(k))$ indicates the mean of the $c$-th node feature. $\delta(\cdot)$ and $\varphi(\cdot)$ represent the ReLU function [20] and Sigmoid function respectively. $W_D$ is the weight set of a convolutional layer with $1 \times 1$ kernels, which reduces the number of channels with the ratio $r$. The channel-upscaling layer $W_U$, where $W_U \bigcup W_D = \mathcal{W}$, increases the channels to its original number with the ratio $r$.

**SA Node Feature Alignment:** The optimization of both offsets and network parameters for local alignment are sensitive to the disturbance of gradients, which makes GAN-based methods perform unstable. Therefore, we minimize the MMD [2, 18] loss to align cross-domain SA node features as:

$$L_{mmd} = \frac{1}{n_s n_s} \sum_{i,j=1}^{n_s} \kappa(\mathbf{h}_i^s, \mathbf{h}_j^s) + \frac{1}{n_s n_t} \sum_{i,j=1}^{n_s,n_t} \kappa(\mathbf{h}_i^s, \mathbf{h}_j^t) + \frac{1}{n_t n_t} \sum_{i,j=1}^{n_t} \kappa(\mathbf{h}_i^t, \mathbf{h}_j^t), \tag{6}$$

where $\kappa$ is a kernel function and we apply the Radial Basis Function (RBF) kernel in our model.

### 3.3 Global Feature Alignment

After having the features $\mathbf{f}_i \in \mathbb{R}^d$ corresponding to the $i$-th sample by a generator network, the global feature alignment attempts to minimize the distance between features across different domains. In difference of local feature alignment, global feature alignment process is more stable due to the invariance of receptive field of inputs, which provides more options for choosing GAN-based methods. In this paper, we apply Maximum Classifier Discrepancy (MCD) [26] for global feature alignment due to its outstanding performance in general-purpose domain alignment.

The encoder $E$ designed for SA node feature extraction is also applied for extracting raw point cloud features: $\tilde{\mathbf{h}}_i = E(\mathbf{x}_i | \Theta_E)$ over the whole object. And the point features are concatenated with interpolated SA-node features as $\hat{\mathbf{h}}_i = [\mathbf{h}_i, \tilde{\mathbf{h}}_i]$ to capture the geometry information in multi-scale. Then, we feed the $\hat{\mathbf{h}}_i$ to the generator network $G$ which is the final convolution layer (*i.e.*, conv4) of PointNet attached with a global max-pooling to achieve high-level global feature $\mathbf{f}_i = max - pooling(G(\hat{\mathbf{h}}_i | \Theta_G))$, where $\mathbf{f}_i \in \mathbb{R}^d$ represents the global feature of the $i$-th sample. And $d$ is usually assigned as 1,024. The global alignment module attempts to align domains with two classifier networks $F_1$ and $F_2$ to keep the discriminative features given the support of source domain decision boundaries. The two classifiers $F_1$ and $F_2$ take the features $\mathbf{f}_i$ and classify them into $K$ classes as $p_j(\mathbf{y_i}|\mathbf{x}_i) = F_j\left(\mathbf{f}_i | \Theta_F^j\right), j = 1, 2$, where $p_j(\mathbf{y_i}|\mathbf{x}_i)$ is the $K$-dimensional probabilistic softmax results of classifiers.

To train the model, the total loss is composed of two parts: the task loss and discrepancy loss. Similar as most UDA methods, the object of task loss is to minimize the empirical risk on source domain $\{X_s, Y_s\}$, which is formulated as follows:

$$L_{cls}(X_s, Y_s) = -\mathbb{E}_{(\mathbf{x}_s, y_s) \sim (X_s, Y_s)} \sum_{k=1}^{K} \mathbb{1}_{[k=y_s]} \log(p((\mathbf{y} = y_s) | G(E(\mathbf{x}_s | \Theta_E) | \Theta_G))). \tag{7}$$

The discrepancy loss is calculated as the $l_1$ distance between the softmax scores of two classifiers:

$$L_{dis}(\mathbf{x}_t) = \mathbb{E}_{\mathbf{x}_t \sim X_t}[\|p_1(\mathbf{y}|\mathbf{x}_t) - p_2(\mathbf{y}|\mathbf{x}_t)\|]. \tag{8}$$

## 3.4 Training Procedure

We apply the Back-Propagation [25] to optimize the whole framework under the end-to-end training scenario. The training process is composed of two steps in total:

**Step1**. Firstly, it is required to train two classifiers $F_1$ and $F_2$ with the discrepancy loss $L_{dis}$ in Eq. (8) and classification loss $L_{cls}$ obtained in Eq. (7). The discrepancy loss, which requires to be maximized, helps gather target features given the support of the source domain. The classification loss is applied to minimize the empirical risk on source domain. The objective function is as follows:

$$\min_{F_1,F_2} L_{cls} - \lambda L_{dis}. \tag{9}$$

**Step2**. In this step, we train the generator $G$, encoder $E$, the node attention network $\mathcal{W}$ and transform network $\mathcal{R}$ by minimizing the discrepancy loss, classification loss and MMD loss to achieve discriminative and domain-invariant features. The objective function in this step is formulated as:

$$\min_{G,E,\mathcal{W},\mathcal{R}} L_{cls} + \lambda L_{dis} + \beta L_{mmd}, \tag{10}$$

where both $\lambda$ and $\beta$ are hyper-parameters which manually assigned as 1.

## 3.5 Theoretical Analysis

In this section, we analyze our method in terms of the $\mathcal{H}\Delta\mathcal{H}$- distance theory [1]. The $\mathcal{H}\Delta\mathcal{H}$-distance is defined as

$$d_{\mathcal{H}\Delta\mathcal{H}}(\mathcal{S},\mathcal{T}) = 2 \sup_{h_1,h_2\in\mathcal{H}} |P_{\mathbf{x}\sim\mathcal{S}}[h_1(\mathbf{x}) \neq h_2(\mathbf{x})] - P_{\mathbf{x}\sim\mathcal{T}}[h_1(\mathbf{x} \neq h_2(\mathbf{x}))]|, \tag{11}$$

which represents the discrepancy between the target and source distributions, $\mathcal{T}$ and $\mathcal{S}$, with regard to the hypothesis class $\mathcal{H}$. According to [1], the error of classifier $h$ on the target domain $\epsilon_{\mathcal{T}}(h)$ can be bounded by the sum of the source domain error $\epsilon_{\mathcal{S}}(h)$, the $\mathcal{H}\Delta\mathcal{H}$- distance and a constant $C$ which is independent of $h$, *i.e.*,

$$\epsilon_{\mathcal{T}}(h) \leq \epsilon_{\mathcal{S}}(h) + \frac{1}{2}d_{\mathcal{H}\Delta\mathcal{H}}(\mathcal{S},\mathcal{T}) + C. \tag{12}$$

The relationship between our method and the $\mathcal{H}\Delta\mathcal{H}$- distance will be discussed in the following. The $\mathcal{H}\Delta\mathcal{H}$- distance can also be denoted as below:

$$d_{\mathcal{H}\Delta\mathcal{H}}(\mathcal{S},\mathcal{T}) = 2 \sup_{h_1,h_2\in\mathcal{H}} \left|\mathbb{E}_{\mathbf{x}\sim\mathcal{S}}\mathbb{1}_{[h_1(\mathbf{x})\neq h_2(\mathbf{x})]} - \mathbb{E}_{\mathbf{x}\sim\mathcal{T}}\mathbb{1}_{[h_1(\mathbf{x})\neq h_2(\mathbf{x})]}\right|. \tag{13}$$

As the term $\mathbb{E}_{\mathbf{x}\sim\mathcal{S}}\mathbb{1}_{[h_1(\mathbf{x})\neq h_2(\mathbf{x})]}$ would be very small if $h_1$ and $h_2$ can classify samples over $\mathcal{S}$ correctly. In our case, $p_1$ and $p_2$ correspond to $h_1$ and $h_2$ respectively, which agree on their predictions on source samples $\mathcal{S}$. As a result, $d_{\mathcal{H}\Delta\mathcal{H}}(\mathcal{S},\mathcal{T})$ can be approximately calculated by $\sup_{h_1,h_2\in\mathcal{H}} \mathbb{E}_{\mathbf{x}\sim\mathcal{T}}\mathbb{1}_{[h_1(\mathbf{x})\neq h_2(\mathbf{x})]}$, which is the supremum of $L_{dis}$ in our problem. If decomposing the hypothesis $h_1$ into $G$ and $F_1$, and $h_2$ into $G$ and $F_2$, and fix $G$, we can get

$$\sup_{h_1,h_2\in\mathcal{H}} \mathbb{E}_{\mathbf{x}\sim\mathcal{T}}\mathbb{1}_{[h_1(\mathbf{x})\neq h_2(\mathbf{x})]} = \sup_{F_1,F_2} \mathbb{E}_{\mathbf{x}\sim\mathcal{T}}\mathbb{1}_{[F_1\circ G(\mathbf{x})\neq F_2\circ G(\mathbf{x})]}. \tag{14}$$

Further, we replace $\sup$ with $\max$, and attempt to minimize (14) with respect to $G$:

$$\min_G \max_{F_1,F_2} \mathbb{E}_{\mathbf{x}\sim\mathcal{T}}\mathbb{1}_{[F_1\circ G(\mathbf{x})\neq F_2\circ G(\mathbf{x})]}. \tag{15}$$

Problem (15) is similar to the problem (9,10) in our method. Consider the discrepancy loss $L_{dis}$, we first train classifiers $F_1$, $F_2$ to maximize $L_{dis}$ on the target domain and next train generator $G$ to minimize $L_{dis}$, which matches with problem (15). Although we also need consider the source loss $L_{cls}$ and MMD loss $L_{mmd}$, we can see from [1] that our method still has a close connection to the $\mathcal{H}\Delta\mathcal{H}$- distance. Thus, by iteratively train $F_1$, $F_2$ and $G$, we can effectively reduce $d_{\mathcal{H}\Delta\mathcal{H}}(\mathcal{S},\mathcal{T})$, and further lead to the better approximate $\epsilon_{\mathcal{T}}(h)$ by $\epsilon_{\mathcal{S}}(h)$.

Table 1: Number of samples in proposed datasets.

| Dataset | | Bathtub | Bed | Bookshelf | Cabinet | Chair | Lamp | Monitor | Plant | Sofa | Table | Total |
|---|---|---|---|---|---|---|---|---|---|---|---|---|
| **M** | Train | 106 | 515 | 572 | 200 | 889 | 124 | 465 | 240 | 680 | 392 | 4, 183 |
| | Test | 50 | 100 | 100 | 86 | 100 | 20 | 100 | 100 | 100 | 100 | 856 |
| **S** | Train | 599 | 167 | 310 | 1, 076 | 4, 612 | 1, 620 | 762 | 158 | 2, 198 | 5, 876 | 17, 378 |
| | Test | 85 | 23 | 50 | 126 | 662 | 232 | 112 | 30 | 330 | 842 | 2, 492 |
| **S\*** | Train | 98 | 329 | 464 | 650 | 2, 578 | 161 | 210 | 88 | 495 | 1, 037 | 6, 110 |
| | Test | 26 | 85 | 146 | 149 | 801 | 41 | 61 | 25 | 134 | 301 | 1, 769 |

## 4 PointDA-10 Dataset

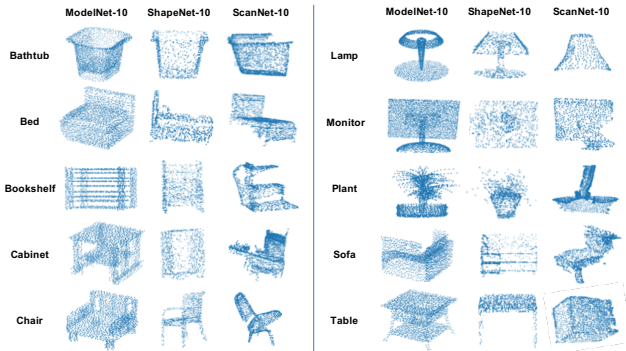

Figure 3: Samples of PointDA-10 dataset.

As there is no 3D point cloud benchmark designed for domain adaptation, we propose three datasets with different characteristics, *i.e.*, ModelNet-10, ShapeNet-10, ScanNet-10, for the evaluation of point cloud DA methods. To build them, we extract the samples in 10 shared classes from ModelNet40 [35], ShapeNet [3] and ScanNet [5] respectively. The statistic and visualization are shown in Table 1 and Fig. 3. Given the access to the three subdatasets, we organize six types of adaptation scenarios which are **M** → **S, M** → **S\*, S** → **M, S** → **S\*, S\*** → **M** and **S\*** → **S** respectively.

**ModelNet-10 (M):** ModelNet40 contains clean 3D CAD models of 40 categories. To extract overlapped classes, we regard 'nightstand' class in ModelNet40 as 'cabinet' class in ModelNet-10, because these two objects almost share the same structure. After getting the CAD model, we sample points on the surface as [23] to fully cover the object.

**ShapeNet-10 (S):** ShapeNetCore contains 3D CAD models of 55 categories gathered from online repositories. ShapeNet contains more samples and its objects have larger variance in structure compared with ModelNet. We apply uniform sampling to collect the points of ShapeNet on surface, which, compared with ModelNet, may lose some marginal points.

**ScanNet-10 (S\*):** ScanNet contains scanned and reconstructed real-world indoor scenes. We isolate 10 classes instances contained in annotated bounding boxes for classification. The objects often lose some parts and get occluded by surroundings. ScanNet is a challenging but realistic domain.

## 5 Experiments

### 5.1 Experiments Setup

In this section, we evaluate the proposed method under the standard protocol [11] of unsupervised domain adaptation on the task of point cloud data classification.

**Implementation Details:** We choose the PointNet [22] as the backbone of Encoder $E$ and Generator $G$ and apply a two-layer multilayer perceptron (MLP) as $F_1$ and $F_2$. The proposed approach is implemented on PyTorch with Adam [15] as the optimizer and a NVIDIA TITAN GPU for training. The learning rate is assigned as 0.0001 under the weight decay 0.0005. All models have been trained for 200 epochs of batch size 64. We extract the SA node features from the third convolution layer (*i.e.*, conv3) for local-level alignment and the number of SA node is assigned as 64.

**Baselines:** We compare the proposed method with a serial of general-purpose UDA methods including: Maximum Mean Discrepancy (**MMD**) [18], Adversarial Discriminative Domain Adaptation (**ADDA**) [32], Domain Adversarial Neural Network (**DANN**) [10], and Maximum Classifier Discrep-

Table 2: Quantitative classification results (%) on PointDA-10 Dataset.

| | G | L | A | P | M→S | M→S* | S→M | S→S* | S*→M | S*→S | Avg |
|---|---|---|---|---|---|---|---|---|---|---|---|
| w/o Adapt | | | | | 42.5 | 22.3 | 39.9 | 23.5 | 34.2 | 46.9 | 34.9 |
| MMD [18] | √ | | | | 57.5 | 27.9 | 40.7 | 26.7 | 47.3 | 54.8 | 42.5 |
| DANN [10] | √ | | | | 58.7 | 29.4 | 42.3 | 30.5 | 48.1 | 56.7 | 44.2 |
| ADDA [32] | √ | | | | 61.0 | 30.5 | 40.4 | 29.3 | 48.9 | 51.1 | 43.5 |
| MCD [26] | √ | | | | 62.0 | 31.0 | 41.4 | 31.3 | 46.8 | 59.3 | 45.3 |
| | √ | √ | | | 62.5 | 31.2 | 41.5 | 31.5 | 46.9 | 59.3 | 45.5 |
| Ours | √ | √ | √ | | 63.7 | 32.1 | 44.5 | 33.7 | 48.2 | 63.0 | 47.5 |
| | √ | √ | √ | √ | **64.2** | **33.0** | **47.6** | **33.9** | **49.1** | **64.1** | **48.7** |
| Supervised | | | | | 90.5 | 53.2 | 86.2 | 53.2 | 86.2 | 90.5 | 76.6 |

Table 3: Class-wise classification results (%) on ModelNet to ShapeNet.

| | G | L | A | P | Bathtub | Bed | Bookshelf | Cabinet | Chair | Lamp | Monitor | Plant | Sofa | Table | Avg |
|---|---|---|---|---|---|---|---|---|---|---|---|---|---|---|---|
| w/o Adapt | | | | | 59.4 | 1.0 | 18.4 | 7.4 | 55.7 | 43.5 | 84.8 | 60.0 | 3.4 | 39.7 | 37.3 |
| MMD [18] | √ | | | | 77.1 | 0.7 | 20.0 | 1.6 | 63.6 | 58.4 | 88.8 | 83.4 | 0.5 | **87.6** | 48.2 |
| DANN [10] | √ | | | | 82.6 | 0.4 | 20.1 | 1.5 | 72.1 | 52.6 | 90.2 | 86.7 | 1.0 | 80.2 | 48.6 |
| ADDA [32] | √ | | | | 84.5 | 1.0 | **22.9** | 2.4 | 66.7 | 62.8 | 83.6 | 70.1 | 1.8 | 86.8 | 48.3 |
| MCD [26] | √ | | | | 84.8 | **4.4** | 18.4 | **7.7** | 74.9 | 62.0 | 85.6 | 80.0 | 1.6 | 82.2 | 50.2 |
| | √ | √ | | | 84.6 | 0.8 | 19.2 | 1.6 | 75.6 | 61.2 | **92.7** | **86.3** | 0.9 | 83.4 | 50.6 |
| Ours | √ | √ | √ | | **85.7** | 2.4 | 20.4 | 1.0 | 79.0 | **64.2** | 90.1 | 83.3 | **3.6** | 83.0 | **51.3** |
| | √ | √ | √ | √ | 84.7 | 1.6 | 19.0 | 1.3 | **81.9** | 63.3 | 90.5 | 82.3 | 2.2 | 82.9 | 51.0 |
| Supervised | | | | | 88.9 | 88.6 | 47.8 | 88.0 | 96.6 | 90.9 | 93.7 | 57.1 | 92.7 | 91.1 | 83.5 |

ancy (**MCD**) [26]. During these experiments, we take the same loss and the same training policy. **w/o Adapt** refers to the model trained only by source samples and **Supervised** means fully supervised method.

**Ablation Study Setup:** To analyze the effects of each module, we introduce the ablation study which is composed of four components: global feature alignment, *i.e.,* **G**, local feature alignment, *i.e.,* **L**, SA node attention ,*i.e.,* **A**, and the self-training [42], *i.e.,* **P**, to finetune the model with 10% pseudo target labels generated from the target samples with the highest softmax scores.

**Evaluation:** Given the labeled samples in source domain and unlabeled samples from target domain for training, all the models would be evaluated on the test set of target domain. All the experiments have been repeated three times and we then report the average top-1 classification accuracy in all tables.

## 5.2 Classification Results on PointDA-10 Dataset

The quantitative results and comparison on PointDA-10 dataset are summarized in Table 2. The proposed methods outperform all the general-purpose baseline methods on all adaptation scenarios. Although the largest domain gap appears on $M \rightarrow S*$ and $S \rightarrow S*$, ours exhibit the large improvement which demonstrates its superiority in aligning different domains. In comparison to the baseline methods, MMD, although defeated by GAN-based methods in 2D vision tasks, is only slightly inferior and even outperforms them in some domain pairs. The phenomenon could be explained as global features limit the upper bound due to its weakness in representing diversified geometry information. In addition, there still exists a great margin between supervised method and DA methods.

The Table 3 represents the class-wise classification results on the domain pair $M \rightarrow S$. Local alignment helps boost the performance on most of the classes, especially for Monitor and Chair. However, some of the objects, *i.e.,* sofa and bed, are quite challenging for recognition under the UDA scenario where the negative transfer happens as the performance could drop on these classes. Moreover, we observed that the imbalanced training samples do affect the performance of our model and other domain adaptation (DA) models, which makes Table 3 slightly noisy. Chair, Table, and Sofa (easily confusing with Bed) cover more than 60% samples in M-to-S scenario which causes the drop of certain classes (e.g., Bed and Sofa).

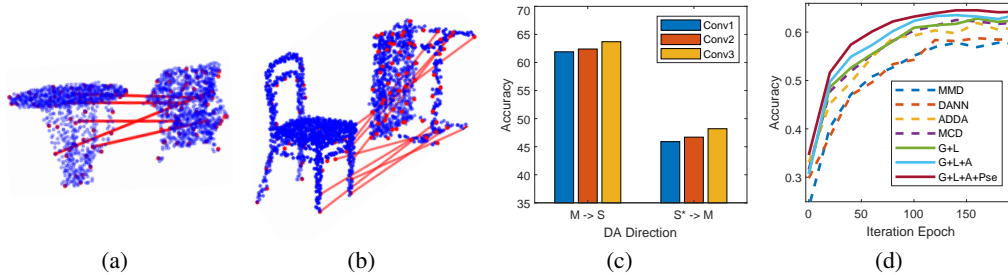

<div align="center">
(a)            (b)            (c)            (d)
</div>

Figure 4: (a)-(b) Matched SA nodes for aligning cross-domain objects. (c) Analysis of different feature extraction layers for local feature alignment, and (d) convergence analysis.

## 5.3 Quantitative Analysis

**Ablation Study:** We further analyze the effect of four components proposed in our model (*i.e.,* $\mathbf{G}$, $\mathbf{L}$, $\mathbf{S}$, $\mathbf{A}$). From the Table 2, we find that together with SA node, adding local alignment will bring significant improvement, but only local alignment with fixed node wouldn't improve a lot. Above results substantially validate the effectiveness of our SA nodes that attributes to its self-adapt in region receptive field and significant weight. And an interesting phenomenon in Table 3 is that the full version method is defeated by $\mathbf{G+L+A}$ in class-wise accuracy. It means that inference of pseudo labels is easily influenced by imbalance distribution of samples in different classes where certain classes would dominate the process of self-training and cause errors accumulation.

**Convergence:** We evaluate the convergence of proposed methods as well as baseline methods on ModelNet-to-ShapeNet in Fig. 4(d). Compared with baselines methods, local alignment helps accelerate the convergence and makes them more stable since being convergent.

**SA Node Feature Extraction Layer:** The influence of different layers for mid-level feature extraction is analyzed in Fig. 4(c) on $\mathbf{M} \rightarrow \mathbf{S}$ and $\mathbf{S^*} \rightarrow \mathbf{M}$. Compared with conv1 and conv2 whose features are less semantical, conv3 contains the best mid-level feature for local alignment.

## 5.4 Results Visualization

We visualize the top contributed SA nodes for local alignment of two cross-domain objects to interpret the effectiveness of local feature alignment in Fig. 4(a)-4(b). The matched nodes are selected from the elements with the highest values from the matrix $\mathbf{M} = \mathbf{h}_i^s \times \mathbf{h}_j^{t^\top} \in \mathbb{R}^{64 \times 64}$ obtained from Eq. 5. It is easily observed that the SA nodes representing similar geometry structure, *i.e.*, legs, plains, contribute most to local alignment no matter they are between same objects or different objects across domains. It significantly demonstrates the common knowledge learned by SA nodes for local alignment.

## 6 Conclusion

In this paper, we propose a novel 3D Unsupervised Domain Adaptation Network on Point Cloud Data (PointDAN). PointDAN is a specifically designed framework based on multi-scale feature alignment. For local feature alignment, we introduce Self-Adaptive (SA) nodes to represent common geometry structure across domains and apply a GAN-based method to align features globally. To evaluate the proposed model, we build a new 3D domain adaptation benchmark. In the experiments, we have demonstrated the superiority of our approach over the state-of-the-art domain adaptation methods.

**Acknowledgements**

We thank Qianqian Ma from Boston University for her helpful theoretical insights and comments for our work.

## Footnotes

[1]The PointDA-10 data and official code are uploaded on `https://github.com/canqin001/PointDAN`

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
