[Reviews · NeurIPS 2019]

Reviewer 1



The authors present the first domain adaptation model for 3D point clouds. They come up with novel structures and modules to create their model. For this, they build on a variety of known and novel techniques, for example they use a PointNet++ encoder, but also introduce novel Self-Adaptive nodes and use a convolution similar to bilateral convolution (they call it deformable convolution) to extract features for domain alignment. The submission seems technically sound and the authors provide a theoretical analysis for their method in terms of the H\Delta H theory. Since I am not an expert in domain adaptation, I did not find conclusive judgement of their contribution on that end. The paper is clearly written, though I noticed a substantial amount of writing mistakes w.r.t. articles (the, a). The presented method achieves clearly better results than other methods undergoing domain transfer without adaptation. It would be interesting, though, to see the result of other methods fine-tuned with a small amount of labelled data to get an impression of the complexity of the domain transfer task between the different datasets. Also, even though there is an ablation study performed for the different proposed parts of the architecture, there is no discussion of the weaknesses of the method, which would be helpful. The approach together with the newly proposed dataset, could be a valuable contribution for the community.

Reviewer 2



# Originality The submission mostly combines the domain adaption loss Maximum Classifier Discrepancy [23] with additional learned local features "Local Feature Alignment" to point cloud classification tasks with unlabeled target domain samples. The driving classification architecture is borrowed from PointNet. The main contribution here lies in the local features that bring up the predictive performance on the target task: small regions, which are centered around sample point cloud points, are first moved with a learned offset (to better support commonalities of the current object) and then weighted by an attention network (to identify important features). The features of these regions are derived from early stages of a PointNet architecture. The final local features are then fed into later layers of a PointNet architecture for classification. The training is done by alternating the training steps from the publication of Maximum Classifier Discrepancy [23]. # Quality The ablation study shows that on average across multiple domain adaptation tasks the added adaptable local features seem to improve over a direct application of general-purpose domain adaptation techniques. However, the effect on different classes seems to vary. # Clarity The description of the architecture and methodology are clear enough. # Significance The contribution -- though successful -- might be of limited significance to the community for mostly two reasons: the derived local feature alignment seems to be mostly a learned weighting and offseting of PointNet features, and the success across classes as shown in table 3 seems noisy; some classes profit from the proposed method (e.g., cabinet) and some don't (e.g., lamp). Minor fixes: - line 25: systems? - line 156, eq 2: Maybe rewriting the equation in the style of an assignment would make sense here? - line 212, eq 10: missing closing parenthesis for h_1(x)? - table 3: MCD and table: Probably remove '1c' here?

Reviewer 3



Originality: - L3: “to the best of our knowledge, there is no method yet to achieve domain adaptation on 3D data, especially point cloud data” see below [SqueezeSegV2: Improved Model Structure and Unsupervised Domain Adaptation for Road-Object Segmentation from a LiDAR Point Cloud, Wu et al 2018] proposes a domain adaptation pipeline for 3D lidar point cloud to reduce distribution gap between synthetic and real data. [Domain Adaptation for Vehicle Detection from Bird’s Eye View LiDAR Point Cloud Data, Saleh et al 2019] This is quite recent, but also explore domain adaptation for synthetic vs real data. Technically the above two are operating in image space (depth semantic segmentation maps, and BEV of point cloud, respectively) but the underlying goal is still to model 3D information from point cloud. The first paper is from 2018 so I do think this paper over claims this ‘first to do domain adaptation in 3D data’ statement a bit. Although it’s worth noting that this paper explores point based representation rather than image based, and for classification task rather than point segmentation. But I think the similarity and different should be mentioned and discussed. + The idea of locally aligning feature using self adaptive node with adaptive part based receptive field is pretty novel and interesting. This provides additional structure to the feature that would make the global alignment easier since it is invariance to scale and part configuration. But while this would work for classification, I’m not sure if it would work for other shape sensitive tasks such as 3D detection? - In fact, Table 3 shows that adding local alignment using self adaptive node doesn’t always lead to an improvement over other baselines on all the class. - Global alignment is a feature alignment from based on MCD from [23], so this part is not new. +/- I think it’s really useful to have a benchmark dataset for domain adaptation, and I appreciate the authors taking initiative and assemble such dataset. But since this is simply a subset of existing data, I don’t think this is a strong contribution (which to be fair, the authors never claim it is) Quality: + Extensive experiments and ablation study with detailed comparison with other UDA baselines. This is really useful especially since the proposed benchmark is new. + The ablation study shows that each component really do add to the performance (Table 2) ? Does the proposed approach with only global alignment equivalent to the MCD baseline from [23]? I assume so since there is no ablation study with only G and G is based on MCD, but are they all using the same setting, parameters, etc? + Performance break down per class in Table 3 is a nice touch. This is very useful since it shows strength and weakness of each approach. - All the scores in Table 3 (Avg) is lower than their Table 2 counterpart, which makes me wonder if the imbalance nature of the data across categories has more effect than it should be. Chair and table sort of dominate the dataset, and skew the final score toward the trend of these two classes. I feel like a more fair comparison is when all class has an equal number of object or when each class is weighted equally. I know that this is pretty common in classification task, but it can be misleading. ? The result for bed is very interesting and worth a discussion. MCD [23] outperform other methods by a large margin. And if we assume that the proposed approach with only G is the same as MCD, then adding local alignment drop the classification score from 26.1 to 4.3 (and adding attention further drop them to 1) Do you have any intuition on why this is the case? Clarity: + Overall the paper is not difficult to understand, + The format of the experiments, ablation study, and the tables to show the results are all very clear and easy to digest. ? I feel like section 3.5 doesn’t add much to the narrative and could be put in the supplementary instead. - It’s not immediately clear to me in line 90-91 that P(s) P(t) refers to the distribution (it’s defined in the next section) Significant: + I believe the idea of self adaptive node for 3D object would be useful to the research community, if it works. Aligning feature might not be new, but doing so in 3D setting and on top of PointNet based feature shows that it is possible and promising, at least for chair and table categories. + It’s true that not much works are looking into Domain Adaptation for 3D data, and it helps to have a common benchmark even if it just a combination of existing dataset. --UPDATED AFTER REBUTTAL-- Thanks for the detailed rebuttal. The additional results are quite interesting and further convince me that the proposed local alignment does help. So I'm keeping my score at 6.

[Author Response · NeurIPS 2019]

We do appreciate all reviewers' careful reviews, constructive comments, and positive recognitions. We firstly answer common questions and then response each reviewer. All the concerns will be addressed in the final version paper.

**Common Questions Response:**

**[CQ1] Class-wise accuracy.** Actually, we observed that the imbalanced training samples do affect the performance of our model and other domain adaptation (DA) models. *Chair*, *Table*, and *Sofa* (easily confusing with *Bed*) cover more than 60% samples in M-to-S scenario which causes the drop of certain classes (*e.g.*, *Bed* and *Sofa*). After re-balancing samples, most classes show satisfactory improvements with our model (seen in the Table below in which each class equally contains 300 samples for training), except for *Lamp*, which indicate the following weakness of our model. 1) Neglect of scale information; 2) When different classes share very similar local structures, our local alignment has the possibility to align similar self-adaptive (SA) nodes across those classes (*e.g.*, large columns contained by both *Lamps* and round *Tables*). We will discuss more details in the final version and explore solutions in our future work.

|  | Bathtub | Bed | Bookshelf | Cabinet | Chair | Keyboard | Lamp | Laptop | Sofa | Table | Avg |
|---|---|---|---|---|---|---|---|---|---|---|---|
| MCD | 74.7 | 28.3 | 10.7 | 16.2 | 75.0 | 99.2 | **83.8** | 99.3 | 5.8 | 76.3 | 56.9 |
| Ours | **79.1** | **37.1** | **12.0** | **30.3** | **75.1** | **100.0** | 79.2 | **99.5** | **10.1** | **77.7** | **60.0** |

**[CQ2] Discuss previous papers.** It is a good suggestion. Our method is similar but different compared with these previous works. [Lai *et al.*, IJRR10] adopts the model trained by Web data to the 3D data scanned by LiDAR for object detection, while it is a supervised fashion which requires labeled target samples for training. In the works of [Wu *et al.*, arXiv18] and [Saleh *et al.*, arXiv19], 3D LiDAR point clouds are firstly projected/transferred into a 2D image and then forwarded to 2D-based conventional algorithms. It is not a general 3D approach which requires specific 3D data and 3D-2D transmission of single view. Our method could directly process point cloud data without any strict limitations.

**[CQ3] Open-source.** We will definitely release our code and all implementation details after the paper is accepted. Moreover, the well arranged dataset will also be released as a solid benchmark to benefit the whole research community.

**Response To Reviewer 1:**

**[Q1] Comparison with bilateral filtering (BF).** Good suggestion. 1) Kernel: BF uses Gaussian Kernel while ours uses a learnable $1 \times 1$ convolutional kernel. 2) Different Objects: BF applies to pixel intensity (RGB) and coordinates to get the weights. Our method applies to high-level features to get the weights. 3) Different weights: In BF, learned weights are for aggregating pixels (*i.e.*, intensity). In ours, learned weights are for aggregating edges (*i.e.*, coordinate).

**[Q2] Analysis of complexity.** Good question. Based on the backbone of PointNet, the parameters of our method is around 12M with 981 MFLOPs/sample. Our method is able to process 1k objects per second on a Nvidia TitanXp GPU.

**[Q3] Mean, stddev and simple size.** Good question. The experiments are repeated over three times with the average one reported on Tab. 2. We further collected stddev in the range of $0.6$-$1.4$, and every object is sampled in 1,024 points.

**[Q4] Results of fine-tuning models.** It's a good suggestion. By selecting 30% labeled target samples in training set, the fine-tuning results on M-to-S and S-to-M are 75.1% and 49.6% respectively. It gains about 4%-5% improvements.

**[Q5 & Q6] Weaknesses** and **Comparison with reference papers.** Please see the answer above **[CQ1]** and **[CQ2]**.

**Response To Reviewer 2:**

**[Q1] Effects of our method on different classes.** It's a good observation. Please see the answer above on **[CQ1]**.

**[Q2] The difference between local alignment and re-weighting of PointNet features.** It's a good question. Local alignment aligns region-level features. Then, aligned region features with the attribute of common domain knowledge are interpolated back to point features, which are updated to bridge the domain gap rather than just re-weighting original PointNet features. In our experiments, the performance has dropped from 69.3% to 62.2% on M-to-S by replacing local alignment with an attention module on PointNet features which indicates re-weighting method doesn't work for DA.

**[Q3] Why some classes profit from DA.** It is a good question. In general, same-class objects, which look similar across domains but discriminative with other class objects, are easier to be aligned. In previous global alignment, the similarity of objects is measured by global semantic features in the feature space. While in 3D data, the global semantic features are often confused between many similar classes, but their geometric local structures could be varied a lot. So in our local alignment module, we try to align the similar local structure representations across domains apart from global alignment, which works especially well for the 3D data containing rich geometric information. For instance, *Cabinets* often have diversified structures, and profit a lot from our SA nodes focusing on unique local structures alignment.

**[Q3] Public Codes & Datasets.** Practical question. Yes, we will do that and please see our reply in **[CQ3]** above.

**[Q4] Minor fixes.** Many thanks for the reviewer's patient reading and careful checking. We do apologize for the inconvenience caused by these problems. We will revise all the typos and correct the formulas in the final version paper.

**Response To Reviewer 3:**

**[Q1 & Q2] Comparisons** and **performance on different classes.** Good observation. Please see reply **[CQ1 & CQ2]**.

**[Q3] Extend to other 3D tasks.** Good question. Classification is the first step and we are planning to extend the method to 3D object detection and segmentation. It is a promising field and we will continuously work in this direction.

**[Q4] Relationship between global alignment and MCD.** Good question. We adopted exactly the same setting for global alignment as those of MCD. Drop of *Bed* is mainly caused by data imbalance detailedly explained in **[CQ1]**.

**[Q5] Section 3.5.** We will provide a more detailed theoretical analysis in the final version and move it to supplementary.

[Meta-Review · NeurIPS 2019]

The submission received mixed ratings prior and after discussion phase. The final scores are 5,6,7 with a tendency of lowering the 7 to a 6. The reviewers acknowledge the quality of the paper, the new dataset and that the technique is a good approach to domain adaption in 3D. The paper is among the first ones, the statements about being the first should be revised given the related literature pointed out by R1. R2 has doubts whether the method is too specificly tuned to PointNet while R3 believes PointNet is an already well enough established method that even if that would be the case the method would still be of interest. The empirical results are varying quite a bit over the classes, the authors answer to this point with a new experiment. This was acknowledged by the reviewers in the discussion and should be added to the final paper. Overall the positive points outweight the doubts about whether the feature alignment is a necessary step to perform 3D domain adaption. The paper presents a valid contribution, with an attempt of a theoretical justification and a novel datasets. Empirical results are sufficient to back the claims made by the paper.